# Review of Materials and Fabrication Methods for Flexible Nano and Micro-Scale Physical and Chemical Property Sensors

**Anesu Nyabadza** [1,2,3,*], **Mercedes Vázquez** [1,2,3], **Shirley Coyle** [3], **Brian Fitzpatrick** [4] and **Dermot Brabazon** [1,2,3]

1   I-Form, Advanced Manufacturing Research Centre, D04 V1W8 Dublin, Ireland; mercedes.vazquez@dcu.ie (M.V.); dermot.brabazon@dcu.ie (D.B.)
2   EPSRC & SFI Centre for Doctoral Training (CDT) in Advanced Metallic Systems, Sheffield S10 2JA, UK
3   Advanced Processing Technology Research Centre, School of Mechanical & Manufacturing Engineering, Electronic Engineering, and Chemical Sciences, Dublin City University, Glasnevin, D09 NA55 Dublin 09, Ireland; shirley.coyle@dcu.ie
4   Oriel Sea Salt Ltd., Clogherhead, A92 V97C Drogheda, Ireland; brianf@orielseasalt.com
*   Correspondence: anesu.nyabadza3@mail.dcu.ie

**Abstract:** The use of flexible sensors has tripled over the last decade due to the increased demand in various fields including health monitoring, food packaging, electronic skins and soft robotics. Flexible sensors have the ability to be bent and stretched during use and can still maintain their electrical and mechanical properties. This gives them an advantage over rigid sensors that lose their sensitivity when subject to bending. Advancements in 3D printing have enabled the development of tailored flexible sensors. Various additive manufacturing methods are being used to develop these sensors including inkjet printing, aerosol jet printing, fused deposition modelling, direct ink writing, selective laser melting and others. Hydrogels have gained much attention in the literature due to their self-healing and shape transforming. Self-healing enables the sensor to recover from damages such as cracks and cuts incurred during use, and this enables the sensor to have a longer operating life and stability. Various polymers are used as substrates on which the sensing material is placed. Polymers including polydimethylsiloxane, Poly(N-isopropylacrylamide) and polyvinyl acetate are extensively used in flexible sensors. The most widely used nanomaterials in flexible sensors are carbon and silver due to their excellent electrical properties. This review gives an overview of various types of flexible sensors (including temperature, pressure and chemical sensors), paying particular attention to the application areas and the corresponding characteristics/properties of interest required for such. Current advances/trends in the field including 3D printing, novel nanomaterials and responsive polymers, and self-healable sensors and wearables will also be discussed in more detail.

**Keywords:** flexible sensors; additive manufacturing; 3D printing; self-healing; nanocomposites; advanced manufacturing



## 1. Introduction

Sensors have been used for over 2000 years [1]. They can be defined as any device that can detect and react to changes in the surroundings. Nowadays, sensors are incorporated in virtually everything and the use of sensors have almost tripled within the last two decades [2,3]. Current sensors enable remote monitoring which allows the transmission of signals to a remote location within a fraction of a second. The advances in real-time monitoring are a huge improvement in comparison to long ago whereby much more time and effort was required to monitor an event. Sensor technologies have definitely improved our way of life. Our smart phones are equipped with a copious number of sensors including sensors that can detect our location, health status, exercise data (e.g., number of steps per day), heart rate and other physiological signals. Currently, smart watches are used to monitor many body signals such as heart rate, temperature, pedometer monitoring

and exercise-related signals. The implementation of flexible sensors into devices allows the creation of a multi-functional device, thereby breaking the limitation of traditional watches. According to Scopus statistics, the number of articles on flexible sensors doubled to 3710 between 2013 and 2019 which shows that the research field is growing rapidly [3]. According to the review paper [3], the market share of flexible displays was 8% in 2016 and rose to 27% by 2020, corresponding with the recent rapid growth of this research area. There are other review papers on flexible sensors published within the last 2 years [3–7], showing rapid growth of this research topic. One report published in 2020 [4] reviewed 3D printed sensors covering force sensors, pressure sensors and others. The report did not capture other manufacturing methods or the use of nanomaterial in depth. Another report published in 2021 [5] concentrated on flexible pressure and strain sensors used in health monitoring. They covered the sensing mechanisms and the use of nanomaterials such as carbon nanotubes in depth, however they did not cover manufacturing methods such as 3D printing which has recently gained much attention for polymer processing. Han et al. [6] discussed the materials, fabrication methods and electrical performance of flexible strain sensors. The sensitivity, linearity, response time and durability properties of the sensors were captured. The materials including flexible polymers and nanomaterials were disused and a short review of 3D printing was presented, however self-healable sensors and the various types of sensors were not presented in detail with the focus only on strain sensors. Gao et al. [7] presented the use of PEDOT:PSS in electrochemical sensors. The review examined sensors that can detect ions, pH levels and hydrogen peroxide. The review was limited/focused on one polymer (PEDOT:PSS) and one type of sensor (chemical sensor). Wen et al. explored various types of sensors and fabrication methods for flexible sensors. The review was centred on applications of the flexible sensors including soft robots but did not present wearable sensing applications and chemical sensors in depth. This review paper presents the most recent developments for the materials and methods used in the fabrication of flexible sensors. This review paper explores the limitations, advantages and advances in current methods and materials, including the use of additive manufacturing (3D printing) in the fabrication of these sensors. Herein, various 3D printing methods being used in the fabrication of various types of flexible sensors including temperature, humidity, pressure, medical monitoring and chemical are explored. A summary of mechanisms and current methods employed in the development of self-healable flexible sensors are presented. Throughout the review, demonstrator examples of the advantages of the flexible sensors are provided. These include their ease of fabrication and increased room for innovative and smart solutions. Some smart sensors that can self-power and change shape upon exposure to a stimuli are mentioned herein. These stimuli-responsive sensors are developed via a new additive manufacturing method called 4D printing.

Sensors are being incorporated in food packaging to enable the detection of cracks in the packaging material and to detect if the food has gone bad. Sensors have enabled the development of smart cars that can fully or partially self-drive. Newer cars can detect road markings and can warn the driver about obstacles such as other cars and trees, which has made driving manoeuvres such as reverse parking much safer and easier.

Sensors can be divided into two main categories, namely non-flexible and flexible sensors. Non-flexible sensors are also termed rigid sensors. Although this type of sensor has pros, such as low cost of substrate material and low power losses, they are limited in flexibility. The lack in flexibility limits their use in health monitoring and other uses whereby the sensor is required to be continually bent, stretched or put under pressure. On the other hand, flexible sensors thrive under deformations. Their electrical properties are not affected by the bending/stretching. In fact, these sensors can use the bending and deformations to detect motion, such is the case of sensors used in robotics and human motion detection. Flexible sensors have several advantages over rigid sensors. They have impeccable sensing capabilities even at harsh bending stresses of 1500 $\mu\varepsilon$ [8]. Some of these sensors can be subjected to 8000 bending cycles and still retain their sensing capabilities [8].



Flexible sensors also tend to have enhanced thermal and mechanical properties and lighter weight than rigid sensors.

The use of flexible sensors is sometimes hindered by the low electrical conductivity of the flexible material. To overcome this, nanomaterials of carbon, silver, copper and others are being incorporated into the flexible component to enhance electrical properties of the device [3]. Polymers are the most used pliable material in flexible sensors due to their high flexibility yet resilience under bending stresses and their ease of fabrication. However, most polymers do not have the required electrical properties for the fabrication of electronics such as health monitoring devices and environmental monitoring devices [3] whereby high sensitivity and conductivity is required. Most efforts in the research of flexible sensors are centred on enhancing the flexibility and conductivity of the materials used in the fabrication.

Areas such as robotics, prosthetics, implantable medical devices, electronic skin and smart watches require flexible and thin sensors. The flexibility of the sensors enables for a simple integration process on curved surfaces such as the neck and wrist. The flexibility property also allows for the development of complex shapes. With shape complexity comes mass, volume and cost reductions. For instance, highly sensitive and recyclable touch sensors made from paper composed of cellulose nanofibrils have been additively manufactured via a cheap method called inkjet printing [9]. The estimated cost of each sensor is EUR 0.06, which demonstrates the cost effectiveness of flexible materials and their ease of fabrication.

Advances in additive manufacturing (AM) has enabled the flexible sensor industry to advance greatly. AM enables complex shapes to be moulded easily by controlling the CAD file. It allows weight reductions and innovative designs to be made that are otherwise impossible with conventional methods, such as moulding and surface coating. With the introduction of various AM methods including powder bed fusion, fused deposition modelling, laser sintering, inkjet printing, aerosol jet printing, stereolithography printing (SLA) and others, various materials can be processed with differing degrees of resolution depending on the intended use. Figure 1 presents some of the recent flexible sensors in literature. These include humidity, chemical, temperature and pressure sensors. Sensors connect humans to the internet enabling the monitoring of phenomena such as the heart status of an elderly person at any time and location, which enables the early detection of emergencies. In Figure 2 we see such an example of the use of flexible sensors in today's digital age [10]. Some researchers are focused on self-healable flexible sensors. Self-healing sensors are able to recover their mechanical and electrical properties after damage such as cracks, scratches or cuts. The ability to self-heal prolongs the life of the sensor which translates to material/cost savings and a decrease in environmental damage due to the disposal of waste products from the manufacturing processes. Self-healing properties enable the sensor to be stable under harsh conditions which translates to a more accurate and reliable sensor and opens up opportunities for innovative uses of the sensor.

Besides flexible versus non-flexible sensor categories, sensors have also been categorised in many ways including intrinsic or extrinsic, active or passive, analogue or digital, absolute or relative, contact type or non-contact type and natural or man-made. One of the most famous ways of categorising sensors is by the stimuli they measure, giving categories such as temperature sensors, humidity sensors, pressure sensors, chemical sensors, light sensor, speed sensor and so on. Sensors can also be named by a special feature they possess, such as self-healable sensors. Recent research in flexible sensors has led to the developments of bio-integrated devices, wearable health monitoring devices and electronic skins, to mention a few. Flexible sensors, coupled with the Internet of Things (IoT), has allowed for remote health care monitoring and human-machine interfaces [11].

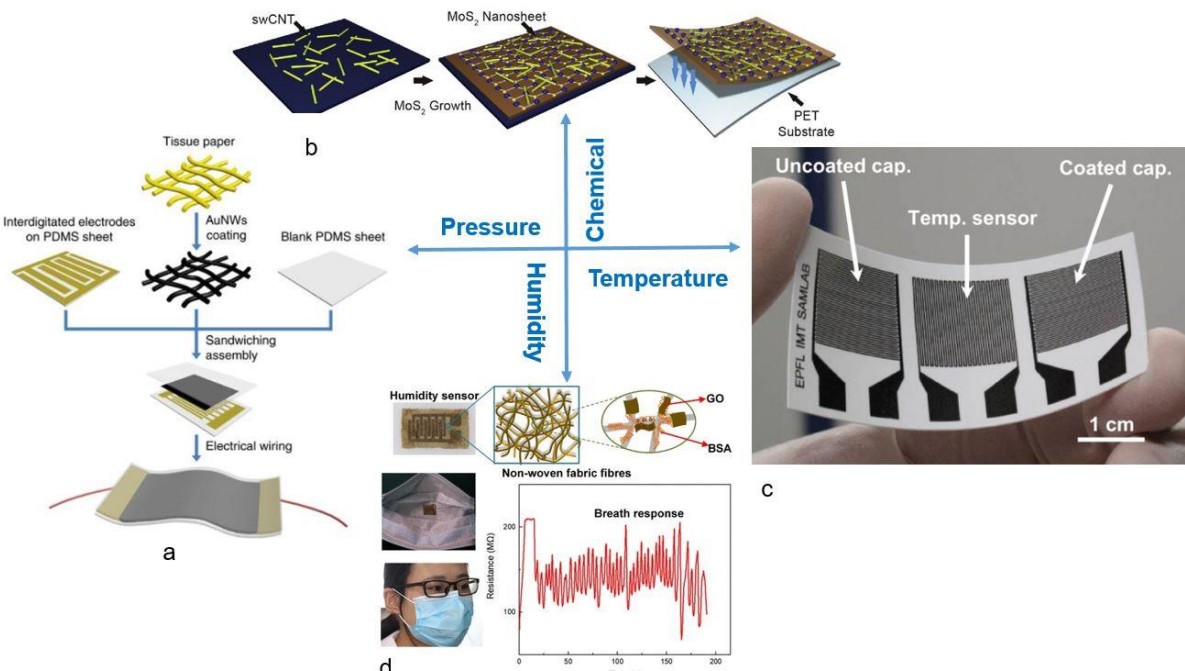

**Figure 1.** A picture of a collection of types of flexible sensors made from a flexible polymer and nanomaterials as the sensing layer. (**a**) Flexible pressure sensor with Au nanorods intertwined with tissue paper and deposited on PDMS polymer, reprinted with permission from [12]. (**b**) Hydrogen gas sensor with carbon/MoS$_2$ nanomaterials on PET substrate, reprinted with permission from [13]. (**c**) Silver nanoparticles inkjet printed on paper to fabricate a flexible temperature sensor, reprinted with permission from [14]. (**d**) Graphene Oxide deposited on fabric for humidity sensing, reprinted with permission from [15].

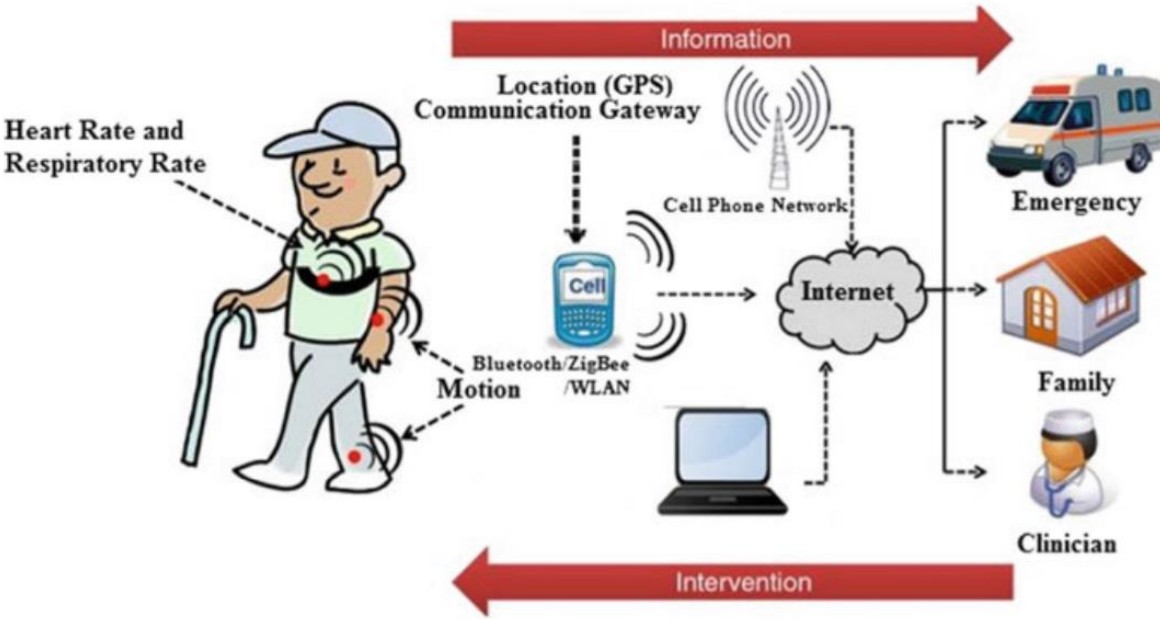

**Figure 2.** Demonstration of everyday uses of flexible sensors. Flexible sensors coupled with the internet allows for the monitoring of things such as heart rate at any location and time. Reprinted with permission from [10].

## 2. Various Types of Flexible Sensors (Temperature, Pressure, Humidity and Chemical)

### 2.1. Overview of Current Materials and Fabrication Methods

Up until the last decade, single-crystal silicon has been the most used substrate in sensor development owing to its distinct advantages including high sensitivity, low power

losses and low cost. Although these rigid silicon sensors have been extensively used, they have certain drawbacks including high stiffness (brittleness), high fabrication cost, high operation power and susceptibility to breakage. Due to the aforementioned drawbacks, flexible sensors have gained much attention in literature and industry [16].

The materials for flexible sensors are processed via various methods depending on the required resolution, size of the sensor and cost. For instance, aerosol jet printing can be used for developing nano/micro sensors at a very high resolution, but the method is highly expensive in comparison to inkjet printing which can also print the same sensors with a lower resolution. Aerosol jet printing is a 3D printing method that involves using metal/non-metal nanoparticle-based inks to print conductive patterns on various substrates, including flexible substrates. The nanoinks are atomised in a chamber and accelerated in a gas (usually nitrogen gas). The ink particles are focused onto the substrate by a nozzle at high speeds to enable a strong attachment of ink particles and substrate. Inkjet printing involves two types of drop-on-demand methods, piezoelectric and thermal. This method has advantages over other printing methods including the lack of warmup time, improved picture quality, good resolution, low cost of printers and ease of implementation. When selecting a fabrication method for flexible sensors, it boils down to a balance among cost, resolution and how easy it is to implement the method. Other methods for the development of micro-sensors include photolithography [17], screen printing [18] and laser cutting [19]; these are discussed in detail in Section 6. In a report by Wang et al., flexible Hall sensors based on graphene nanoparticles and Kapton were developed via photolithography [17]. These sensors were calibrated with a commercial Hall sensor (Allegro MicroSystems A1324) to give normalised sensitivity values. The graphene sensors were found to have sensitivity values of orders of magnitudes higher than the commercial sensor. The sensors have a thickness of 50 μm, a minimum bending radius of 4 mm and a voltage and current normalised sensitivity of up to 0.093 $VVT^{-1}$ and 75 $VAT^{-1}$, respectively. These values of sensitivity are comparable to a silicon-based rigid sensor which have voltage and current sensitivities of 0.1 $VVT^{-1}$ and 100 $VAT^{-1}$ respectively. No degradation of the flexible sensors was found after 1000 bending cycles with bending radius of 5 mm, making the sensors compatible in uses such as biomedicine and wearable electronics, where the precise monitoring of current and position are imperative. When fabricating wearable devices, strain, pressure and temperature sensors are among the most important components. The current research area involves enhancing the materials used in fabricating these sensors. These include nanomaterials such as graphene [11,17], carbon nanoparticles, MXene [20], cellulose nanocrystals [20], copper nanowire [21], silicon nanomembranes [11], cellulose nanofibrils [9], silver nanomaterials [3,22], gold nanoparticles [3], nickel nanoparticles [23] and polymers, including polyimide, Kapton [17], polyglycerol sebacate [24], polyethylene glycol, polyvinyl alcohol [25], polydimethylsiloxane (PDMS), polyurethane (PU), polyethylene terephthalate and hydrogels [25]. Various colloidal nanocrystal-based sensors have been developed including pressure, temperature and strain gauge sensors [11].

### 2.2. Humidity Sensors

Flexible humidity sensors with a graphene oxide (GO) sensing layer deposited on piezoelectric ZnO thin film on polyimide substrate have been fabricated [8]. A total of 4 μL of the GO of various concentrations were drop cast on the surface of the devices and left to evaporate to obtain GO thin films. These flexible sensors were bent 8000 times and remained operable, demonstrating their excellent flexibility and reliability for industrial applications. Another humidity sensor based on fabric and graphene oxide was developed in 2020 for respiration monitoring to support healthcare delivery in response to the COVID-19 pandemic [15]. The researchers claim that the fabric material ranks above polymer materials when fabricating flexible sensors due to the poor hygroscopicity and breathability of polymers, which limits their comfortability and sensitivity. Before this, in 2016, another research team developed a humidity sensor used in respiratory disease monitoring composed of biodegradable paper infused with carbon nanotubes [26]. The

sensor is easily attached under the person's nose and can measure respiratory related signals (airflow rate, humidity and temperature) which can be correlated to respiratory disease (see Figure 3). The sensor measures the real time humidity, temperature and airflow rate of the patient by correlating changes in resistance due to heat coming from the person's breath. The sensor is reported to be eco-friendly and cheap, paper and carbon being the raw materials.

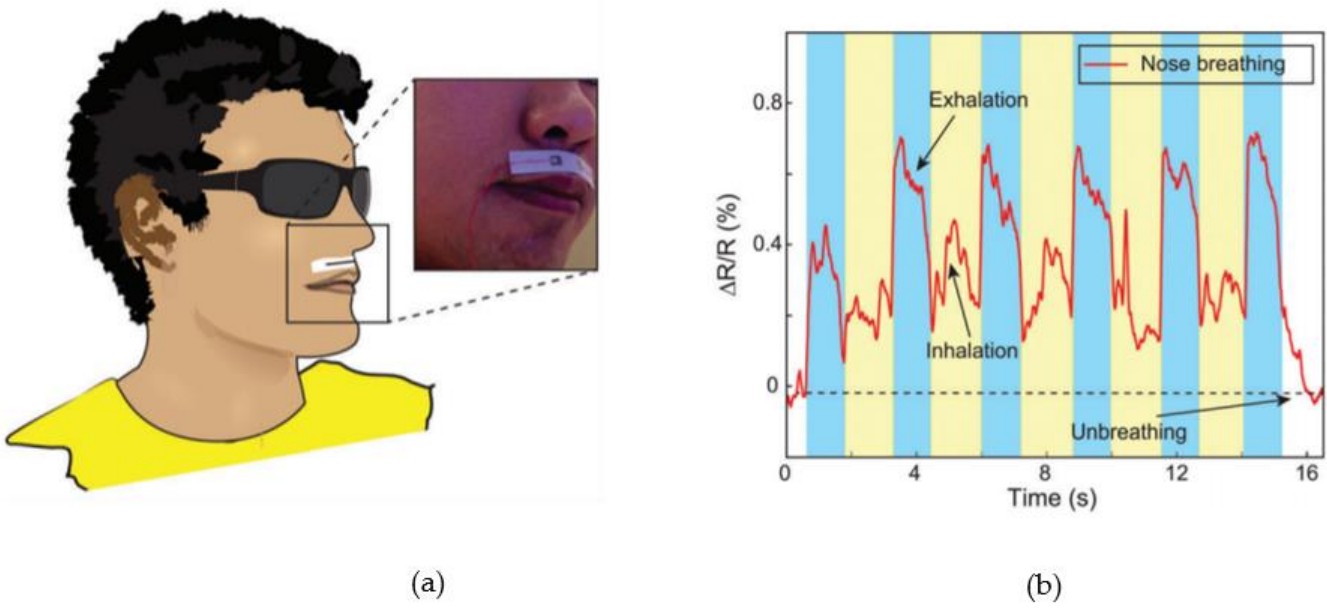

(a)  (b)

**Figure 3.** A picture of a patient wearing a flexible respiratory disease monitoring sensor (humidity sensor), reprinted with permission from [26].(**a**) Flexible respiratory sensor attached to a person's nose for monitoring.(**b**) Output from the flexible respiratory sensor showing distinct peaks for inhalation and exhalation.

### 2.3. Pressure Sensors

Pressure sensors have gained attention in flexible sensor research due to their versatility. These sensors detect the forces exerted on a surface area. Pressure sensors are used in various applications including medical diagnosis and touch screen devices. The most cost effective and efficient type of pressure sensors are the capacitive and resistive types, which convert force/pressure into electrical signals [27,28]. Current research in pressure sensors is centred on improving the flexibility, stretchability, durability, linearity and sensitivity of the sensors [11,12,29].

Currently, most touch sensors are rigid and are made from substrates such as glass or thermosets and sensing materials such as microelectrodes or indium tin oxide [9]. These materials are not recyclable and are bad for the environment. Microelectrodes require expensive disposal methods at the end of their use. A research team from the South China University of Technology developed highly sensitive touch sensors which are biodegradable. The sensors are composed of nanopaper made from cellulose nanofibrils with dimensions of 3.5 cm × 2.5 cm × 0.05 mm [9]. Currently, there exist three major issues in flexible sensor research, namely reliability under bending cycles, sensitivity and transparency. All three issues were addressed by the research team during the development of paper-based sensors. In addition to developing usable and reliable sensors, the team mentioned that each sensor would cost about EUR 0.06, which is cost effective. The low cost comes from the fact that cellulose, the main component of the touch sensor, is the most abundant biopolymer on earth.

The availability of cellulose makes cellulose paper a cheap and sustainable substitute for microelectrodes. The only issue with cellulose is its low transparency, which was addressed by the research team by using cellulose nanofibrils composed of cellulose [9]. Nanomaterials tend to have better physicochemical and plasmonic effects than their bulk

counterparts. Cellulose nanofibrils have a high transparency of 80% at 400 nm, a value much higher than the bulk counterpart. These sensors can withstand 1000 bending cycles at a minimum bending radius of 5 mm without significant degradation or loss in sensing capabilities. The research team demonstrated another advantage of flexible sensors which is their ease of fabrication. A simple inkjet printing process was used to fabricate the capacitive sensors. Capacitive sensors have an advantage over resistive and inferred-sensing techniques including low power requirements, high reliability and ease of fabrication [30].

Biocompatibility and disposal after use remains an issue in the flexible sensor research. The research by Ling et al. tackles these issues, developing touch sensors made from paper which are non-toxic to the human body and can be disposed safely in the environment with a decomposition period of 3–4 weeks when placed into soil [9].

Nanocrystal-based strain gauge sensors are used in fabricating wearable devices for monitoring heartbeat, breathing rhythms, human motion and others [11]. Strain sensors can measure the electromechanical deformation of objects. Nanomaterial-based strain sensors have advantages over the conventional metal thin film-based sensors, including higher sensing capacity and flexibility. This enables nano-based sensors to be used in advanced cases such as modelling the human body by measuring the curvature of the human arm and in voice recognition by measuring the resistance of the sensors with respect to the movement of the vocal cords, thereby distinguishing the words spoken [31].

### 2.4. Temperature Sensors

Temperature is an important indicator in many industries including food storage, air conditioning control and aviation. In some cases, slight deviations in temperature can mean something significant, therefore the sensitivity, response times, accuracy and reliability of the sensor needs to be at high standards. Temperature measurements can be used to indicate the critical status in a manufacturing plant, the stability of a car engine or the health of a human being.

Flexible temperature sensors are required in body temperature monitoring devices. These are useful especially for health monitoring and gained much interest during the COVID-19 pandemic whereby smart masks and skin temperature sensors have been deployed for early detection of the virus [32]. The accuracy and sensitivity of the temperature sensor is imperative, especially in medical applications.

The human body often indicates sickness by a deviation in the normal temperature of 36–37.5 °C. Temperature measurement has been used by doctors for centuries as a means to detect illness. During the COVID-19 pandemic, the use of temperature sensors has increased rapidly because one of the major symptoms of disease is an increase in body temperature to above normal levels. During the pandemic, in many countries, people must pass a temperature test before entering a shop or a bus. Some workplaces in the Republic of Ireland have a temperature sensor on the entrance that can alert you if your temperature is above normal levels, which could be a sign for the COVID-19 disease. Many schools worldwide are measuring the temperature of the students daily via inferred temperature sensors. Inferred sensors are being used to allow social distancing as these sensors can detect the temperature from a distance. An alternative and more efficient way to provide real time temperature data is the use of flexible temperature sensors attached to the skin or embedded into the clothing or face mask. The main challenges in fabricating these types of sensors are achieving comfort for the user, bendability/durability of the sensor after bending cycles, interference with water (washability), achieving good flexibility, achieving required electrical properties and printing related issues. In the following sections, various examples from the literature are examined, in which researchers are tackling the aforementioned issues in various innovative ways.

### 2.4.1. Working Mechanism

Temperature sensors operate based on a material property that varies with temperature. This could be resistance, volume or light properties. The changes in environmental

temperature triggers changes in the property of interest (e.g., resistance). The changes in the property are measured and mathematically correlated to changes in temperature. There exist several types of temperature sensors and, among these, the most popular are resistance temperature sensors (RTS), thermistor and thermocouple. Thermocouple sensors are fabricated using two different metals such as copper/constantan, iron/constantan or chromel/alumel. Thermocouples work on the basis of the Seebeck effect [33]. Thermistors contain materials that vary in resistance with changes in temperature. This change can be measured and calibrated to temperature measurements. Thermistors can measure small variations in temperature, and they come in various shapes and sizes. Common thermistor shapes are bead, rod and disk shapes. The bead type is the smallest in size with a diameter of $\leq 1.25$ mm [34]. For construction of RTSs, silver, copper, gold and nickel are the most used materials [35]. Recently, carbon-nanomaterials and conductive polymers have gained much attention in the fabrication of RTSs [2,26,36,37].

### 2.4.2. Related Work

In a report published in 2020, a skin conformable GO/PEDOT: PSS based temperature sensor that can sense a temperature range of 25–100° was fabricated [38]. The sensor was subjected to 100 bending cycles and its sensitivity remained excellent. The sensor was attached to a robotic arm and used to control the robot as per temperature changes. This demonstrates the real-life application of sensors and their importance in robotics and electronic skins. Humidity interference is a major concern in temperature sensors and researchers are looking for ways to tackle this.

A textile temperature sensor that can constantly measure body temperatures was fabricated by Husain et al. [39]. The sensor was fabricated on an industrial scale flat-bed knitting machine whereby a metal-based sensing layer was embedded into the knitted layers, as shown in Figure 4. The working mechanism of the sensor is that the metal layer changes its resistance in response to temperature changes. Various metal wires were tested and modelled including nickel, copper and tungsten, with the most promising among these being nickel and tungsten owing to their high availability, sensitivity and high reference resistance. Another textile temperature sensor based on graphene-coated polypropylene textile fibres was reported [40]. The unique selling points of this sensor was its low voltage requirements (1 V) and its washability. The sensor was tested for washability by placing it in a glass beaker filled with water at various temperatures (30–50°C) with detergents of various types and spinning at various speeds (400, 800 and 1000 rpm) for an hour. SEM was used for morphological analysis. The sensor was subjected to 1000 bending tests at a bending radius of 5 mm and survived the mechanical deformation without significant decrease in sensing ability. Another study by Dankoco et al. used inkjet printing to develop a flexible resistive-based temperature sensor based on silver ink on Kapton substrate [41]. The sensor can sense temperatures between 20–60 °C with a voltage range of 0–1 V. This temperature range makes the sensor a good candidate for human temperature sensing.

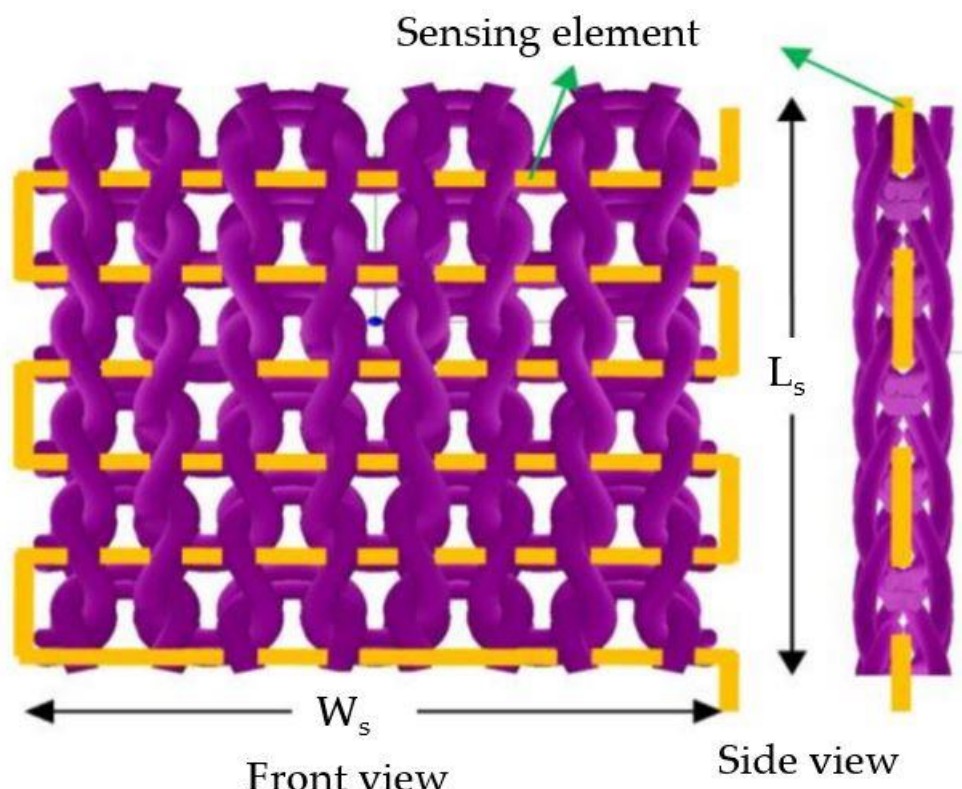

**Figure 4.** A textile temperature sensor, metal rods are intertwined with the fabric, reprinted with permission from [39].

### 2.5. Chemical Sensors

Chemical sensors can detect a certain chemical element or compound (the analyte) within a sample which may be in gas or liquid form. In 2014, a research team developed an electrochemical ionic sensor based on single-walled carbon nanotubes (SWCNTs) with sodium n-dodecyl sulfate (SDS), an anionic surfactant [42]. The sensor has potential use in flexible electronics and the authors demonstrated its ease of fabrication via a simple inkjet printing of colloidal inks on flexible material using a commercial desktop printer (Hewlett-Packard (HP) inkjet printer). High-accuracy gas detection has become an essential property for preventing environmental pollution caused by Nitrogen Dioxide ($NO_2$), Ammonia ($NH_3$), Carbon Monoxide (CO), Carbon Dioxide ($CO_2$), fine dust, etc. This makes the monitoring of these gases essential for preventing pollutant emissions accompanying industrial and recycling processes. Recent reports have used transition metal dichalcogenides (TMDs) in flexible chemical sensors for increasing the sensitivity. TDMs examples include molybdenum disulfide ($MoS_2$), MXenes (2D carbides/nitrides), molybdenum diselenide ($MoSe_2$), tungsten disulfide ($WS_2$), tungsten diselenide ($WSe_2$), borophene (2D boron), silicene (2D silicon), germanene (2D germanium) and hexagonal boron nitride (h-BN) [43]. TDMs have a high surface-area-to-volume ratio which provides more reaction sites for the reaction with specific molecules [44]. In another study, $MoS_2$-SWCNT nanosheets were used to fabricate a chemical sensor that can detect $NH_3$ and $NO_2$ gases due to the excellent gas adsorption property of SWCNTs [13]. The $MoS_2$ nanosheets were fabricated via a chemical vapor deposition (CVD) technique using a porphyrin-type organic promoter. The team reported that the sensors exhibited stable chemical sensing with low loss in sensitivity even after $10^5$ bending cycles.

In another report, a hydrogen gas flexible sensor was fabricated from (Tungsten trioxide) $WO_3$ nanowires grown on Kapton via an aerosol-assisted chemical vapour deposition method [45]. The sensors could measure various concentrations of hydrogen gas. Silver and silver-platinum were used to coat the sensors. The sensors were then subjected to

bending cycles and it was found that the silver-platinum coating reduced the flexibility of the sensor, rendering it unusable, while the silver coating was successful after cyclic bending tests.

In a review paper on recent research in flexible sensors based on nanomaterial that was published in 2020 [3], the uses of flexible sensors were explored including biomedicine, smart devices, environmental monitoring and automobile manufacturing. Practical examples of uses of flexible sensors were mentioned, including sensors for monitoring glucose [46] and pulse [47]. Table 1 summarises examples of temperature, pressure, humidity and chemical flexible sensors from the literature.

**Table 1.** Examples of types of flexible sensors (temperature, pressure, humidity and chemical), the materials used in their fabrication and their potential uses.

| Sensor Type | Materials | Potential Applications |
|---|---|---|
| Temperature | 1. PDMS and graphene nanowalls (GNWs)<br>2. Cellulose and graphene oxide<br>3. PDMS and graphene oxide<br>4. Parylene and silver nanoparticles<br>5. Kapton and silver nanoparticles<br>6. PDMS, chromel and alumel<br>7. PEDOT:PSS and carbon nanoparticles<br>8. PEDOT:PSS, graphene oxide and silver<br>9. Polypropylene and graphene | Monitoring body temperature. [48]<br>Electronics. [49]<br>Electronic skin. [50]<br>Environmental sensing. [14]<br>Monitoring body temperature. [41]<br>Microactuators. [33]<br>Skin temperature sensing. [36]<br>Robotics. [38]<br>Clothing. [40] |
| Pressure | 1. PDMS and graphene oxide<br>2. Cellulose and MXene<br>3. Silicon and AlGaN/GaN<br>4. Silicon nitride and graphene oxide<br>5. Tissue paper, PDMS and Au nanorods<br>6. Silicon and PDMS<br>7. Airlaid Paper (AP) and Carbon Black<br>8. PEDOT:PSS and PDMS<br>9. Silk and graphene | Electronic skin. [50]<br>Wearables. [20]<br>Wearables. [27]<br>Wearables. [28]<br>Wearables. [12]<br>Electronic skin. [29]<br>Healthcare/wearables. [47]<br>Wearables. [51]<br>Clothing/skin sensing. [37] |
| Humidity | 1. PDMS, ZnO and graphene oxide<br>2. Parylene and silver nanoparticles<br>3. Fabric and graphene oxide<br>4. PET, Au nanoparticles and graphene oxide | Flexible electronics. [8]<br>Environmental sensing. [14]<br>Respiration Monitoring. [15]<br>Environmental sensing. [52] |
| Chemical | 1. Sodium n-dodecyl sulfate and SWCNTs<br>2. $MoS_2$ and SWCNTs<br>3. Kapton and Ag/Pt and $WO_3$ nanowires | Electrochemical sensing. [42]<br>$NH_3$ and $NO_2$ gas sensing. [13]<br>$H_2$ gas sensing. [45] |

## 3. Nanomaterials

Nanomaterials have gained much attention in the flexible sensor industry. There exists two main ways of fabricating flexible sensors with nanomaterials incorporated. One of the ways involves the inclusion of nanoparticles/nanorods/nanowires within a polymer matrix. The polymer provides flexibility while the nanomaterial provides conductivity or a sensing property. The advantage of this strategy is that multiple types of nanomaterials can be included within the polymer matrix which enables advanced sensing properties, innovations and high sensitivity [53,54]. The second way involves the incorporation of a low Young's modulus laminar conductive material on a flexible substrate. The nano-sized laminar has better conductive properties than nanoparticles/nanorods/nanowires due to the lack of insulating material within the laminar structure (increased surface area without polymer matrix in contact).

Nanomaterials have exceptional electrical properties compared to their bulk counterparts [55,56]. This makes them a prime candidate in flexible sensor technology whereby the device's weight and volume must be minimised while its electrical properties and bendability needs to be maximised. Nanomaterials are often incorporated within the

flexible substrate matrix to induce electrical properties into the device. Nanomaterials of carbon, copper, iron, gold, magnesium and others have been used in the fabrication of flexible sensors. The market share of conductive inks based on nanomaterials has risen in the last decade and is predicted to continue rising. The report here [57] shows the market share of conductive inks in China alone; carbon-based inks are the most used followed by silver-based nanoinks, mainly due to the lower costs, attractive electrical properties and biocompatibility.

*Carbon Nanomaterials and Others*

Carbon nanomaterials are the most used conductive materials in flexible sensor technology. Carbon nanomaterials have excellent conductive and mechanical properties and cost effectiveness but have poor transparency, which is one the of the main challenges in this research field. Another challenge in using carbon-based nanomaterials such as carbon nanotubes is that the particles tend to agglomerate due to the hydrophobic nature of carbon nanotubes and strong van der Waals forces between them. Agglomeration causes print head/cartridge nozzle clogging during inkjet printing of carbon inks. This can be counteracted by the use of organic solvents such as dimethylformamide or N-methyl2-pyrrolidone or the use of dispersants for nanotubes in water solutions. Carbon particles in water can also exhibit high surface tension leading to printing issues. This can be dealt with by the use of a surfactant [42]. Another carbon-based nanomaterial currently used in flexible sensor technology is graphene. Graphene is a semi-metal with zero bandwidth and is widely used as a sensing layer in flexible sensors. The resistance of graphene is inversely proportional to temperature changes, making it a good candidate for temperature sensors. Graphene oxide and its reduced version has been used as a sensing layer, with reduced graphene oxide having a higher conductivity and therefore a wider usage [49].

In one report, an electrochemical sensor was fabricated using single-walled carbon nanotubes (SWCNT) with sodium n-dodecyl sulfate (surfactant) [42]. The sensor was fabricated by inkjet printing the SWCNT ink using a Hewlett-Packard (HP) inkjet printer onto a thin film. The resulting sensor had a competitive sheet resistance of 132 $\Omega$. The electrical properties of the sensor were characterised via cyclic voltammetry (CV) which involves the cycling of the electrodes voltage and measuring the output current. Another interesting research area is the use of flexible sensors on human skin. The main research challenge is that human skin does not perfectly differentiate between pressure and temperature stimuli under mixed stimulation. A research study by Bae et al. overcame this limitation by developing a sensor that can not only sense both temperature and pressure at the same time but can also differentiate between them [50]. The reduced graphene oxide-based sensor exhibited linear sensitivity to both pressure and temperature variations. The sensor has a pressure sensitivity of 0.7 $kPa^{-1}$ up to 25 kPa and a temperature coefficient of resistance of 0.83% $K^{-1}$. The pressure/temperature sensor can sense temperatures in the range of 22–70° and has a competitive response time of 100 ms. Both pressure and temperature sensing correlated to resistance changes of the GO sheet.

A highly sensitive and wearable temperature sensor was fabricated using graphene nanowalls combined with polydimethylsiloxane (PDMS) (see Figure 5a) [48]. Some researchers have incorporated nanoparticles of multiple types in polymer matrixes or on their surface to fabricate advanced materials for flexible sensors. For example, Zhang and his team [58] developed a strain sensor based on PDMS with silver nanoparticles and carbon nanotubes on its surface. The silver/carbon nanocolloid was drop cast onto the PDMS substrate and the resulting composite was dried to evaporate the liquid, resulting in a stretchable sensor that can attain various shapes, as shown in Figure 6b. The drop casting fabrication method is shown in Figure 6a.

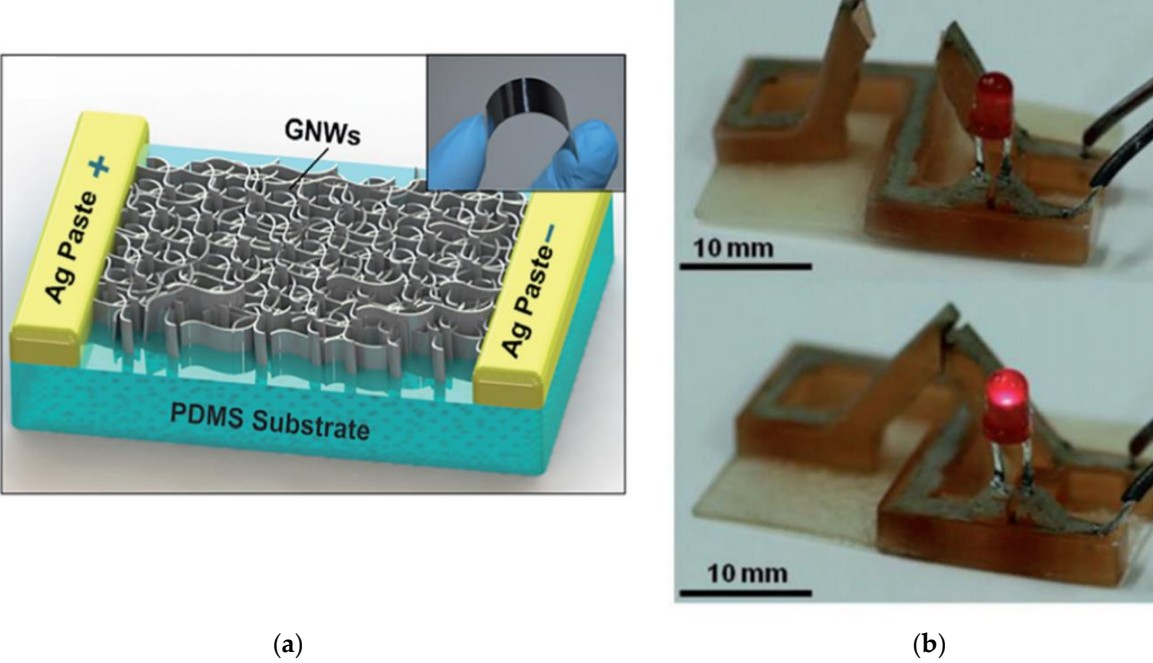

(**a**)　　　　　　　　　　　　　　　(**b**)

**Figure 5.** Flexible sensors based on nanomaterial and a polymer. (**a**) PDMS/Ag nanoparticles on PDMS layer, reprinted with permission from [48]. (**b**) Carbon nanomaterials on a shape memory polymer capable of detecting temperature changes by opening/closing an LED circuit. The sensor changes shape when the temperatures rises/falls, thereby switching a circuit on/off, reprinted with permission from [59].

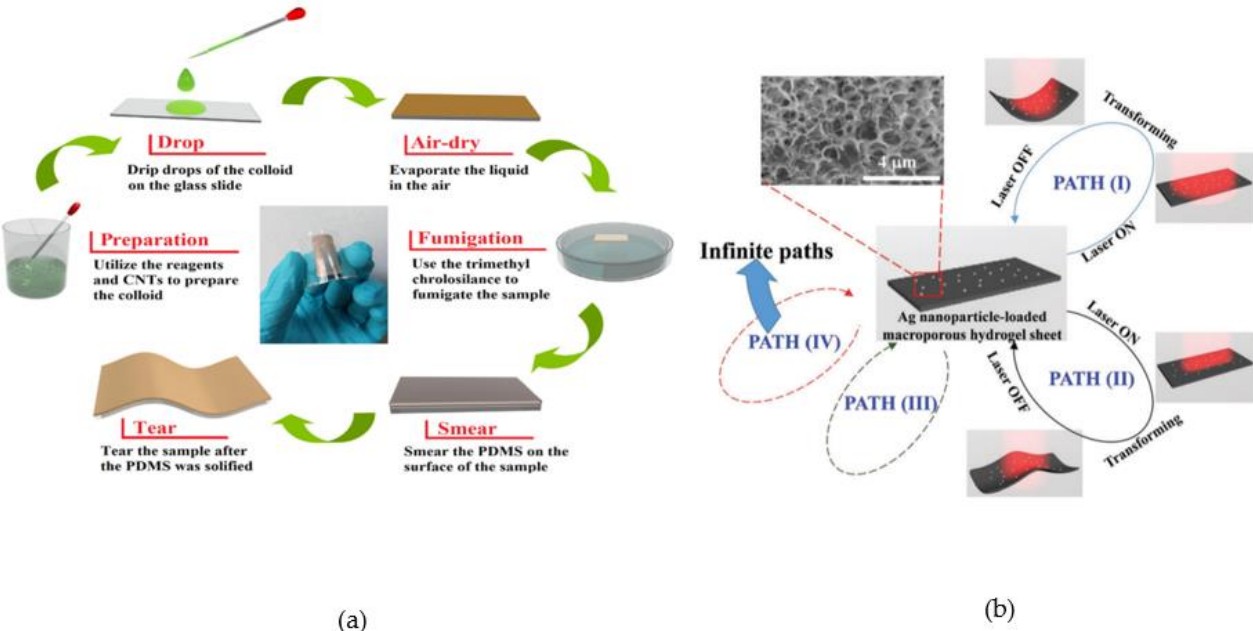

(a)　　　　　　　　　　　　　　　(b)

**Figure 6.** A shape transforming sensor triggered by light, reprinted with permission from [58]. (**a**) Drop-casting fabrication method, nanocolloid is deposited onto a flexible substrate. (**b**) Different shape transformations of the sensor depending on the position on which the sensor is exposed to light.

Recently, a sensor that is responsive to an external magnetic field was fabricated [60]. The sensor is composed of magnetic nanoparticles, Iron Oxide ($Fe_3O_4$), embedded into alginate and methylcellulose hydrogel. The hydrogel brings unique properties into the composite, including high stretchability, softness and biocompatibility, which enables the

sensor to have versatile uses including stimuli-responsive soft robotics, agricultural and biomedical actuators and tissue engineering. The research team used the CELLINK BIO X 3D printer (CELLINK, Sweden) to print $10 \times 10$ mm cubes with various infill density (10–75%). The nanoparticle concentration was varied from 7.5–15% $w/w$. It was found that an increase in nanoparticle concentration increased the responsiveness (measured as bending angle towards the magnet) of the fabricated sensor when subjected to a rectangular-shaped neodymium magnet ($20 \times 50 \times 5$ mm), as shown in Figure 7. The bend angle of the sensor towards the magnet increased with decreasing infill (lower amount of hydrogel). The sensor has potential for remote actuation and can be made into various shapes by means of 3D printing. Various analytical techniques, including TEM, SEM, XRD, attenuated total reflectance-Fourier transform infrared (ATR-FTIR), were used to analyse the nanoparticle size/morphology/chemical composition. Magnetic measurements were performed using a MPMS-XL (Quantum Design) SQUID magnetometer. Other characterisation techniques included compression tests, viscosity measurements and a thermal stability test. All the aforementioned characterisation techniques gave insights to the stability, usability and robustness of the sensor in a real-world application.

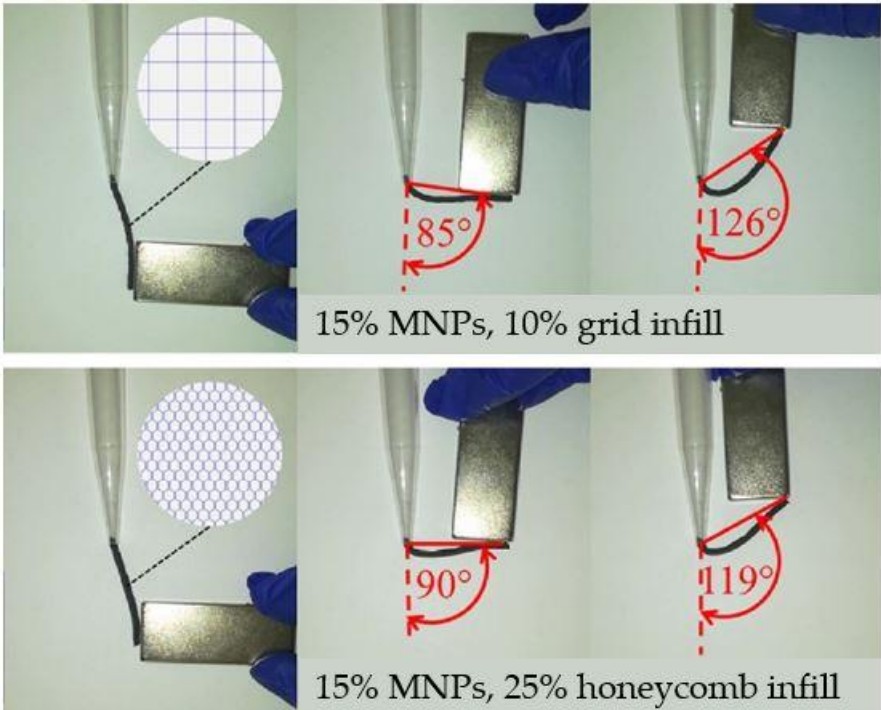

**Figure 7.** A magnetically actuated sensor based on flexible hydrogel (alginate and methylcellulose hydrogel) and magnetic nanoparticles (MNPs). The responsiveness of the sensor (measured as bending angle towards the magnet) is proportional to the concentration of the MNPs, reprinted with permission from [60].

Gold nanomaterials have good mechanical properties and have better electrical conductivity than carbon nanomaterials. However, gold is much more expensive than carbon and its superiority in electrical properties usually does not always justify its massive cost in comparison to carbon. In a published paper, Gold nanowires (Au NWs) impregnated in a tissue paper were sandwiched with PDMS films to fabricate a flexible pressure sensor with a fast response time of less than 17 ms [3]. The sensor can detect pressures down to 13 Pa, has a high sensitivity of over 1.14 KPa$^{-1}$ and has a high stability of over 50,000 loading–unloading cycle. In another report, a silver nanoparticle-based temperature sensor was fabricated via inkjet printing on paper. The sensor can sense temperatures in the range of $-20$ to $60^\circ$ with excellent linearity [14]. Similarly, spherical silver nanoparticles impregnated in a poly(N-isopropylacrylamide) (PNIPAM) matrix were used to fabricate a light

sensor that can transform to multiple shapes depending on the light pattern [61]. In another report, Au nanorods incorporated in PNIPAM matrix were used to develop a light/thermal responsive sensor with competitive response speed and potential for remote actuation [62]. Similarly, gold nanoparticles were used to fabricate electrodes that were incorporated into a PNIPAM bi-layer to develop a flexible sensor with temperature and pH responsivities [63].

## 4. Polymers

Various polymers are being used in the fabrication of flexible sensors due to their low weight, adjustable electrochemical properties, flexibility, comparatively low cost and ability to be processed in solution. The next section gives an overview of some of the most widely used polymers in literature and industry.

### 4.1. Polydimethylsiloxane (PDMS)

PDMS has gained much attention as a material in the fabrication of flexible sensors due to its biocompatibility, transparency, non-flammability, non-toxicity, hydrophobic nature and stretchability. It is formed from repeating units of siloxane monomers. It is the most used substrate material for flexible sensors with rheological requirements. PDMS is used as the flexible material and as a cover for the conductive component within the flexible sensor. PDMS bonds strongly with nanomaterials which enables the fabrication of nanocomposites with enhanced electrical properties. In one report, PDMS with SWCNTs was used to fabricate a pressure sensor capable of sensing small pressures such as two small insects [64]. The sensor has excellent transparency, high sensitivity and a fast response time. PMDS has been extensively used as a substrate in the fabrication of flexible sensors with the incorporation of nanomaterials. It has been used together with silver nanowires to fabricate flexible wearable heaters [65]. PDMS as a substrate is compatible with various types of nanomaterials in the fabrication of flexible sensors including carbon black [66], carbon nanotubes [67], Au–TiO2 NWs [68] and Ag nanoparticles [69].

Although PDMS has valuable properties in flexible sensors, it is limited in adhesion between itself and the conductive layer which reduces the sensitivity of the sensor. An innovative way to get around this issue is to premix the PDMS and the conductive nano-material before fabricating the sensor. Another way to enhance PDMS adhesive properties is by oxidising the surface functional groups thereby changing the surface from a poor adhesive hydrophobic surface to a good adhesive hydrophilic one. Methods of oxidising the surface include exposing it to UV light and subjecting it to oxygen plasma [51].

The structure of the PDMS also influences its sensitivity properties. It has been proven that PDMS films with a microstructure have higher sensitivity and shorter response times than those without a microstructure. Microstructure studies give room for further investigations in improving the microstructure of the PDMS films in order to optimise the sensitivity and response times of the sensors. A research team in the Suzhou Institute of Nano-Tech and Nano-Bionics managed to control the microstructure of PDMS films by using silk as a template and changing its vertical and horizontal portions [64].

### 4.2. Poly(N-Isopropylacrylamide) (PNIPAM)

Poly(N-isopropylacrylamide) (PNIPAM) is a temperature responsive polymer, making it a good candidate for temperature sensors. Its low density of 1.1 g/cm$^3$ and flexibility have led to its use in flexible sensors [61,70,71]. The main challenge when using PNIPAM in sensors is to achieve fast responses and integrity during shape changes. To counteract this, researchers have been incorporating nanomaterials of graphene oxide [70], silver [61], gold [62,72] and others [71] to develop new advanced composites. These composites have better physicochemical properties than the isolated components. The added nanomaterials increase conductivity thereby increasing the sensitivity of the sensor and, in some cases, the added particles introduce a new functionality such as light or pH responsiveness. Such is the case in a study by Zhang et al. [62], when gold nanorods were incorporated into the polymer matrix via electrospinning to develop a composite that is not only temperature

responsive but also light responsive due to the surface plasmon resonance of gold nanorods. The added nanomaterials also increased the sensitivity of the sensor. The sensor could then increase its temperature from room temperature to 34.5 °C in 1 s and further increase this temperature to 60 °C in 5 s during laser irradiation. This sensitivity upgrade is due to the remarkable property of gold nanomaterials being able to absorb near-infrared (NIR) irradiation and turn it into heat energy in a short period of time. This heat energy is then used to heat up the thermoresponsive polymer to achieve a quick response. The research team further concluded that these Au/PNIPAM composite-based sensors can be easily mass produced due to the high feed rate of 0.6 mL/h and high productivity rate of over 20 cm$^2$/h via electrospinning.

In another report, PNIPAM impregnated with silver nanoparticles was used to fabricate a light responsive sensor capable of transforming into many shapes, including boat-like, hoof-like, and helical and saddle-like structures [61]. The research team claim that the sensor can transform to an infinite number of shapes depending on the light pattern (see Figure 6b). The sensor can revert to its original flat shape upon removal of the light source. The sensor responds to near-infrared (NIR) light. NIR light is preferable to UV or visible light as an actuation method because it is less harmful. A laser of wavelength 808 nm was used to irradiate the sensor for actuation and depending on the light irradiation pattern. A unique feature of this sensor was the introduction of pores (200 nm–1 μm) which led to the reduction in response time of the sensor. The pores increased the rate of water molecules transport when the polymer undergoes swelling or shrinkage thereby increasing the speed of shape transformation. The speed of shape transformation was also increased by the introduction of the silver nanoparticles. Silver nanoparticles can absorb NIR light and can turn it into heat energy quickly. The heat is then used to locally heat the polymer thereby inducing shape changes when the temperature goes above 32 °C, the lower critical solution temperature of PNIPAM. The pores were fabricated by the addition of poly-(ethylene glycol) (pore-creating reagent) to the hydrogel and followed by UV photopolymerisation and washing of the composite thereafter to remove the PEG, leaving sub-micron pores. The advantages of this sensor are its competitive response time, potential for remote actuation and the low actuation temperature (32 °C). Its disadvantage is the need for a constant stimulus for a transformed shape to remain. However, the sensor can be used in normally open switching mechanisms.

### 4.3. Other Polymers

Other substrates used in the fabrication of flexible sensors include polyimide (PI), polyethylene Naphthalate (PEN), poly(3,4-ethylene dioxythiophene), polyethylene terephthalate (PET) and polystyrene sulfonic acid (PEDOT:PSS). PET is developed from ethylene glycol and dimethyl terephthalate. It is semi-crystalline in nature and has a higher viscosity than PDMS. PET is used to fabricate common plastic objects including plastic bottles and packaging material. PEN is a polymer similar to PET, but it has higher dimensional and temperature stability. It is derived from ethylene glycol and carboxylate polymer. PEN is commonly used as a solar cell protection material. Kapton has also been used as a substrate in the development of temperature sensors [41].

### 5. Additively Manufactured Flexible Sensors (3D Printing)

Additive manufacturing (AM), also called 3D printing, has many advantages over other manufacturing methods, such as subtractive manufacturing, dye casting and moulding. These advantages include its flexibility in design and material savings. The field of AM has grown so much that there now exists a copious amount of AM methods including fused deposition modelling (FDM), selective laser melting (SLM), electron beam melting (EBM), aerosol jetting, inkjet printing and many others. The development of new AM methods has allowed various materials to be processed, including polymers, metallic nanoinks, ceramics, nanocomposites and alloys. Various AM methods are being used to produce flexible sensors. Extrusion methods such as FDM/FFF are the most used AM methods for

processing polymers. FDM involves the melting and extrusion of a polymer filament onto the print bed in a layer-by-layer fashion to build the part according to the CAD file. FFF has the same format except that the filament in FFF is usually custom made to produce certain mechanical/electrical/chemical properties in the filament before the printing process. During FFF of flexible sensors, the molten polymer filament is usually mixed with 56 conductive nanomaterials, such as graphene or silver nanoparticles, before printing the sensor according to the CAD file. By controlling the polymer-nanomaterial ratio, the CAD file (printing pattern) and the printing parameters (print speed, bed temperature, etc.), one can directly control the properties of the printed sensor such as resistivity, sensitivity, size and rigidity. This enables the development of complex shaped and tailored sensors, which is otherwise difficult to achieve with conventional methods.

Many examples of 3D printed sensors exist in the literature; the field has grown very much in the last two decades due to the increased availability and declining costs of 3D printers. One can now purchase a standard FDM desktop 3D printer online at under EUR 500, and the cost is predicted to continue falling due to the increased number of 3D printer manufacturers. An interesting example of the use of AM is an inductive sensor that was 3D printed via coaxial extrusion method [73]. The sensor was fabricated by extruding silicon rubber and gallium–indium alloy liquid at the same time. The sensor was installed on a human finger and could capture different degrees of bending. A similar example involved an FDM 3D printer that was used to print structures which were then combined with liquid metal to pattern conductive patterns with microstructures [74]. Flexible sensors can be easily fabricated by combining 3D printed flexible structures with basic micro resistors, capacitors and inductors.

Commercial polymer filaments for FDM include acrylonitrile butadiene styrene (ABS), polylactic acid (and versions of these with carbon nanoparticles incorporated), polycarbonate (PC), PU, polyphenylsulfone, poly (ether ether ketone) and poly(ether imide)s. In one report, FDM was used in fabricating capacitive and piezoresistive sensors composed of polycaprolactone infused with carbon black to induce electrical properties [75]. First, the polymer and the carbon black were mixed and moulded into a filament of 1.5 mm diameter to match the requirements of a desktop 3D printer. The resulting filament was then extruded and printed to form a sensor that exhibited piezoresistive properties. Another research team developed a polymer nanocomposite composed of polyurethane infused with MWCNTs. FDM was then used to print the nanocomposite into piezoresistive strain sensors [76].

The printing of multiple materials simultaneously to fabricate a multi-material object has gained much attention in the literature. This is because multi-material objects enable the fabrication of objects that have versatile properties. For instance, a flexible polymer and a conductive material can be printed simultaneously to produce a multi-material object with the flexibility properties of the polymer and the conductive properties of the conductive material. Multi-material printing has enabled the growth of 4D printing technologies whereby the printed part exhibits some intelligent or smart feature such as shape memory or colour changes upon exposure to certain stimuli [56]. An FDM printer that can print two materials simultaneously was used to process a polyurethane filament in one nozzle and a carbon nanotube composite filament in another nozzle to fabricate a flexible force sensor [77]. The ability to print multiple materials saves printing time in comparison to one-material printers. It also enables innovation and more complexity in design (e.g., micro channels with conductive material within a polymer object.)

Although FDM has several advantages including the low cost of printers and materials, it has two disadvantages, namely the requirement of filaments of a certain diameter and a certain range of melting points and it can only process materials with a certain range of rheological properties. On the other hand, direct ink writing (DIW) printing method overcomes these limitations. Various polymers, conductive nanofillers, elastomers, hydrogels and nanocolloids can all be printed via this method, demonstrating its superior versatility over FDM. The material can be printed at room temperature in the form of

droplets, filaments or aerosols in building the part in a layer-by-layer fashion. The inks for DIW are categorised as metal, MXenes and conductive polymer composites. The conductive properties of the printed objects depend on the electron transfer ability of the printed ink. Nanoparticle inks of carbon and silver are the most prevalent in flexible sensor literature, owing to their high conductive properties, stability after printing and low cost (carbon inks). Metal nanoinks (e.g., copper, titanium, magnesium and iron) are seldom used because, although their conductive properties are superior to those of carbon/silver, they are easily oxidised in the presence of moisture which makes them lose their electrical capabilities, rendering them unusable in this case. In addition, metal leakage could be detrimental to the environment. Strain sensors based on flexible PDMS and silver nanoinks have been successfully fabricated via DIW [78].

MWCNTs have been used in flexible sensor technology. MWCNTs have the ability to entangle nicely through hydrogen bonding with polymers such as PVP and PVA, which enables the carbon nanomaterials to be well dispersed and integrated into the polymer matrix. This allows for the development of conductive polymer composites that can be used in FDM, FFF and DIW 3D printing. In another report, MWCNTs were dispersed in a chitosan matrix with citric acid, acetic acid and lactic acid and used to fabricate a strain sensor [79]. The sensor has self-healing capabilities and is water driven, enabling it to have a long operating life and potential to be self-powering.

The high fabrication cost of flexible sensors had been the major concern until the introduction of additive manufacturing into the field. Nowadays, customised sensors are built rapidly with ease thanks to 3D printing given how straightforward it is to manipulate a CAD file in comparison to traditional methods such as moulding or dye casting. Force sensors are extensively used in fields such as robotics and health monitoring whereby the forces detected by the sensor are translated to information relating to the robot/human movements. Many force sensors have been fabricated via 3D printing in literature [4,80].

Other materials that have gained incredible interest in the field of flexible sensors are conductive nanoinks (nanocolloids), owing to their superior electrical and plasmonic properties over solid materials such as wires. The inks have an added advantage because they can be processed via various AM methods, including cheap and easy to implement methods such as inkjet printing. Standard desktop inkjet printers can be used to print conductive patterns, even on simple substrates such as paper. The main challenge in inkjet printing conductive nanocolloids is getting the ink to a particular range of viscosity values acceptable by the printer. Highly viscous organic liquids such as glycerol have been used in pursuit to control the viscosity of the conductive ink, however care has to be taken such that the glycerol does not inhibit the conductive properties of the ink. In another report, six functional inks, including a piezoelectric and a conductive ink, were used in fabricating soft strain gauge sensors within micro-channels [74]. These sensors have potential use in the fields of toxicology and drug screening. In another study, nanoinks of carbon, silver and manganese dioxide were inkjet printed on cellulose-based paper to fabricate supercapacitors and other sensors [81]. Kapton was also used as a substrate in the inkjet printing of silver nanoinks to develop a sensor [41].

Another innovative flexible sensor fabricating technique involves using an AM method such as SLA to 3D print flexible devices from polymers which are then coated with a conductive ink of silver/carbon nanoparticles to introduce electrical properties. The polymers can have shape memory effects which makes the device intelligent and self-powering. For example, flexible temperature sensors were fabricated by coating a 3D printed shape memory polymer with a silver nanoink via a sintering technique at room temperature [59]. The shape of the sensor changes upon increasing/decreasing the temperature. The sensor is shown in Figure 5b. It is shaped in such a way that when the temperature is increased its shape changes from the top shape in Figure 5b to the bottom shape, enabling for a conductive path to be created. The sensor can be used in detecting temperature changes by opening/closing a circuit. It can be used as a thermometer or as a safety feature at a production plant or in a device.

Polymers with shape memory behaviour enable the fabrication of intelligent devices that can self-power and be controlled remotely. The ability to self-power further reduces the volume and bulkiness of the device which opens new application opportunities for sensors whereby weight reduction is imperative (e.g., the aerospace industry). A shape memory sensor based on polyurethane and carbon black was printed via FDM [82]. The device's shape memory effect is triggered by sunlight which enables it to be self-powering. The device has potential use in temperature sensing and weather monitoring. A review of current state of the art shape memory nanocomposite devices that are currently used in various fields, including heath monitoring, energy storage/harvesting and sensing technologies, was published in early 2021 [56]. This review touches on the area of 4D printing, an interesting topic at the moment.

In another report, a 3D printed flexible glove was embedded with a temperature and other sensors to create a smart wearable medical device which is tailored to the condition of the patient [83]. The glove has conductive channels that accommodate a heater, sensors, actuators, resistors, antennas and capacitors, generating a smart, flexible and tailored medical device. The glove is used for comfort thermotherapy of the patient's hand. The heat produced by the electronics could help treat injured areas by increasing blood-flow and relieving pain. Similarly, a flexible prosthetic hand made from Tango plus (a flexible material) was 3D printed using a commercial 3D PolyJet printer (Eden260V) [84]. A flexible temperature sensor was embedded onto the surface of the prosthetic via hydrogen bonding. The prosthetic hand is capable of retaining its original electrical properties at various hand signals. The 3D printed hand could sense the temperature of a human hand accurately.

Additive manufacturing (AM) has gained the attention of the flexible sensor industry, owing to its main advantages of flexible design, repeatability, wide range of printable materials (as we have seen in this report), ability to print multi-material items, reduced cost and customised micro/nano structures. AM has been used to fabricate various tactile sensors that can detect stimuli such as shear, bending, torsion, vibration frequencies reaching 400 Hz and pressures in the range of 5 Pa–100 Kpa [85]. In one report, a flexible strain sensor was fabricated by embedding silver nanowires between two PDMS layers [58]. The sensor showed great piezoresistivity with variable gauge factor and high stretchability reaching 70%. Piezoelectric effect involves the conversion of mechanical force to electrical signals and has been widely used in the fabrication of flexible sensors. Traditionally, ceramic-based piezoelectric materials have been used for sensing technologies, however these are limited in flexibility and cannot be used were flexibility and bendability are required, such as wearable and implantable medical devices. Hence, polymer-based piezoelectric materials such as poly(vinylidene fluoride) and copolymerized poly(vinylidene fluoride-trifluoroethylene) have gained attention in the fabrication of piezoelectric sensors [86]. A piezoresistive tactile sensor was fabricated via a PolyJet-based 3D printing machine (Connex 500, Stratasys Objet Co., Ltd., Rehovot, Isreal) [87]. A commercially available flexible material called TangoPlus was used in the fabrication process. The sensor consists of two layers of the TangoPlus sandwiching MWCNTs piezoresistive lines. Another research team used a photopolymerisation-based 3D printer to print a flexible material with micro-channels (diameter of 500 microns) into which conductive nano silver was placed [88]. Initially the channels were filled with wax, which was then melted away to place the conductive silver nanomaterials, which acted as a piezoresistive sensing part.

Laser beam techniques, such as SLM, have also been used in fabricating flexible sensors as in the case in one report whereby metal powder was used as the 3D printing material to fabricate helical-shaped electrochemical electrodes of various sizes [89]. These sensors exhibit pH sensing, oxygen catalytic properties and good capacitive properties. Three-dimensional printed sensors and their characteristics are summarised in Table 2. Some general advantages and disadvantages of commonly used 3D printing methods in flexible sensor fabrication are shown in Table 3.

**Table 2.** Examples of additively manufactured (3D printed) sensors and some key characteristics studied in literature.

| Printing Method | Type of Sensor | Stability/Minimum Bending Cycles | Sensitivity (Smallest Detectable Quantity) |
|---|---|---|---|
| Coaxial extrusion. [73] | Inductive sensor | 500 bending/stretching cycles | 0.001–0.25 µH/mm |
| FDM. [74] | Inductor–capacitor-resonant tank circuitry for monitoring the quality of liquid food. | n/a | 4.3% resonance frequency shift |
| FDM. [75] | Capacitive and piezoresistive sensors | n/a | n/a |
| FDM. [77] | Multiaxial force sensor | 1 000 bending cycles | ~2.11 N/mm |
| FDM. [80] | Force sensor | 38 MPa Young's modulus | n/a |
| FDM. [82] | Environmental monitoring | 30 °C Tg | 76 mW/cm$^2$ |
| FDM. [83] | Wearable (programmable heater, temperature sensor and circuitry) | 0–80 °C | n/a |
| FDM. [85] | Tactile sensors | 5 Pa–100 Kpa | n/a |
| FDM. [84] | Wearable (temperature sensor) | | ~0.225 kΩ/°C |
| DIW. [78] | Strain sensors | 1–30% stain | |
| DIW. [79] | Strain sensor | Strain at break of 180% | |
| Inkjet printing. [81] | Supercapacitors | 3 000 bending cycles | 300Ω/sq sheet resistance, power density 96 mW/cm$^3$ |
| Inkjet printing. [41] | Temperature sensor | 20–60 °C | 2.23 × 10−3/°C |
| Stereolithography. [59] | Temperature sensor | ~27–~39 °C | >98% strain fixity rate, >93% strain recovery rate |
| Photopolymerisation. [88] | Piezoresistive sensor | 5.5 MPa Young's modulus, elongation at break of 18.3% | n/a |
| SLM. [89] | pH sensing | n/a | n/a |
| Directprint/cure (DPC) and projection-based stereolithography. [87] | Piezoresistive tactile sensor | n/a | n/a |

**Table 3.** General advantages and disadvantages of commonly used 3D printing methods in flexible sensor fabrication.

| 3D Printing Technique | General Advantages | General Disadvantages |
|---|---|---|
| FDM/FFF | <ul><li>Cheap materials</li><li>Wide ranges of printers from cheap to expensive depending on needs</li><li>Fast printing</li><li>Easy material customisation (e.g., adding nanomaterial into polymer matrix)</li><li>Print speed can be varied depending on required quality</li><li>Portability</li><li>Easy to use</li></ul> | <ul><li>Limited resolution</li><li>Limited to polymers</li></ul> |
| Inkjet printing | <ul><li>Higher resolution than FDM/FFF</li><li>Accurate printing</li><li>Cheaper than aerosol jetting</li><li>Wide range of inkjet printers</li><li>Well known technique</li><li>Nanoink printing</li><li>Portability</li></ul> | <ul><li>Limited in substrate materials</li><li>Specific rheology requirements</li><li>Print head clogging</li></ul> |

**Table 3.** *Cont.*

| 3D Printing Technique | General Advantages | General Disadvantages |
|---|---|---|
| Stereolithography (SLA) | • Higher resolution than FDM/FFF<br>• Potential for multi-material printing<br>• Accurate printing | • Limited to UV curable materials<br>• Printers are more expensive that FDM or inkjet printers |
| Aerosol jetting | • Higher resolution than Inkjet printing, FDM, Stereolithography and DIW<br>• Substrates can be polymers, ceramics or metallic<br>• Multi-material printing<br>• Curved surface printing<br>• Nano/microelectronics printing | • Expensive |
| Direct ink writing (DIW) | • Higher resolution than FDM<br>• Multi-material printing<br>• Easy material customisation (e.g., adding nanomaterial into polymer matrix)<br>• Portability | • Limited to low melting point materials |

## 6. Self-Healing Flexible Sensors

Self-healing implies the sensor can repair itself after damages incurred during use from torsion forces, cuts, cracks, fractures, curling, bending, friction forces, scratching and other damages. The ability to self-heal improves the service lifespan of the sensor as well as enhance its performance by reducing losses in sensitivity due to damages (scratches, cracks, cuts, etc.). The ability of a sensor to recover itself from damage reduces service costs and reduces the use of materials in producing more sensors. Sensors are incorporated in virtually all aspects of our lives, as shown in Figures 1 and 2 (e.g., in food packaging and health monitoring), and the production of sensors is predicted to continue rising. Therefore, the service lifespan of these sensors should ideally be long to avoid contamination of the environment due to increased sensor-waste disposal. Self-healing can be achieved in a number of ways. One of the ways is whereby a repair material is embedded within the sensor such that in the event of a crack, the repair material is released due to the expansion of the crack. The repair material would either simply fill the gap itself or react with a catalyst and polymerise to fill the gap. In either case, this method is usually irreversible. The second method to achieve self-healing is whereby the bonds within the polymer can rebind after a crack spontaneously or under the action of an external stimulus such as heat or light. When a material can change shape or self-heal after being 3D printed, it is called a 4D printed object, with the fourth dimension being time.

There exist two main ways of fabricating self-healing sensors. One way involves the use of a self-healing polymer as the flexible material with a conductive material layer. Another way involves premixing the self-healing polymer with the conductive nanomaterial to produce a self-healing nanocomposite that can then be used to fabricate the sensor via a particular method (e.g., moulding and 3D printing).

In a study by Liu et al., a self-healing $CaCu_3Ti_4O_{12}$ sheet with SWCNTs on the top and bottom surfaces was used to fabricate a capacitive sensor [37]. The sensor was subject to a crack and was able to recover its original capacitive value and tensile property after subjection to heat at 150 °C for half an hour [37]. Similarly, a self-healing ammonia gas sensor that could self-heal within 30 min was prepared by incorporating MWCNTs within polyelectrolyte multilayer matrix [90]. In another report, SWCNTs were incorporated into a polymer matrix with self-healing properties and heat sensitivity in producing a temperature sensor [84]. The sensor can recover from damages such as cracks upon heating. The polymer matrix shifts upon heating which moves the conductive network thereby recreating the electrical connection.

Other than biocompatibility, flexibility and good transparency, PDMS also has self-healing properties. Two self-healing PDMS sheets were used to cover a silver nanoparticle based conductive layer in fabricating a sensor for human–machine interaction. The sensor

exhibited good electrical properties with a conductivity value of 714 $Scm^{-1}$ [91]. The self-healing properties of PDMS increases the service life of the sensor.

Human skin comprises of about 7% of total body weight and covers 100% of the human body surface [92]. The skin has amazing abilities including self-healing and sensing stimuli such as pressure (touch) and temperature (heat). The research of flexible electronic skin has recently gained much attention in fields such as robotics, health monitoring and human–machine interactions. In any case, the flexible electronic skins need to respond to electrical signals translated from physical phenomenon such as touch or temperature changes. There exist two challenges in flexible electronics skin research. One of the challenges is achieving the self-healing property of the skin and the second challenge is the ability to distinguish directions, just as the actual skin. Self-healing ability enables a stable operation and increases the life span of the system while the ability to distinguish stimuli directions enables an intelligence of the electronic skin for practical uses [93]. A research team developed an electronic skin sensor based on carbon nanotubes (CNTs), polyurethane and epoxidized natural rubber [93]. The electronic skin can detect human motion, has self-healing properties and has the ability to distinguish direction [93]. The sensor was made from a hydrogel produced by cross-linking PVA and PEI with Bn, and MXene was incorporated into the matrix to enhance electrical properties (PVA/Bn/PEI/MXene (PBPM)). The PBPM electronic skin can self-heal within 0.06 s, which is excellent in comparison to other reported hydrogel-based electronic skins. The skin also has an impressive response time of 0.12 s due to the interaction of the copious functional groups (e.g., –OH and –O) with the polymer matrix. The electronic skin can distinguish the direction of stimuli, such as wrist up/down and head up/down. The research team claims to be the first to report such a direction recognition ability in flexible electronic skin literature. The self-healing ability of the skin can be attributed to its supramolecular interactions and the dynamic covalent bonds that revive its electrical and mechanical properties upon damage. During the fabrication of the electronic skin, samples were characterised via SEM for morphology studies and X-ray photoelectron spectroscopy (XPS) for surface analysis and chemical composition. Functional groups were characterised via Fourier transform infrared spectroscopy (FT-IR). Electrical signals from tensile stain stimuli were captured by an electrochemical workstation. Tensile strain signals were measured at a deformation rate of 100 mm min$^{-1}$.

Recyclability remains an issue in the fabrication of flexible electronic skins. Hydrogels are often used in the construction of these skins due to their ability to self-heal, however most of these skins can be re-used up to 3–5 times [94]. During recycling, electrical conductivity of the electronic skin can decrease slightly due to evaporation of a small amount of water in the hydrogel. Hydrogels can be easily moulded into various shapes which brings flexibility in design. A PVA/Bn/PEI/MXene electronic sensor showed impeccable recyclability of up to 30 cycles without significant degradation in mechanical properties and or need for treatment [93].

A self-healing polymer matrix composed of dynamic Diels–Alder (DA) adducts with $CaCu_3Ti_4O_{12}$ (S-CCTO) nanoparticles incorporated into the polymer matrix were used in fabricating a self-healing motion sensor [95]. The motion sensor can self-heal from a cut by a blade within 30 min by heating to a temperature of 105 °C. $CaCu_3Ti_4O_{12}$ has gained considerable attention in literature owing to its huge dielectric permittivity and thermal range from 100 to 500 K [96]. The sensor has potential use in rehabilitation, sports performance measurements and in the entertainment industry. These sensors can be incorporated in clothing or on human skin, offering real time monitoring. Naturally, the sensor is subject to bending, stretching and cuts, therefore a self-healing element is imperative to ensure reliability and reduction in safety hazards. Furthermore, a finger motion sensor was developed using the same materials for the electrodes which were then spray-coated on all surfaces with SWCNTs [95]. The SWCNTs showed homogeneity on the surface of the electrode, according to SEM and TEM measurements, which led to the required conductivity of the electrodes. After being subjected to damage, the self-healing composite layer moves, which leads to the separated SWCNTs to re-join and construct

conductive paths. The electrode was placed in an LED circuit to test its electrical and self-healing abilities. The electrode allowed enough current to pass through to light up the LED which demonstrates it has functional conductive properties. The electrode was then cut with a blade, leaving a micro-gap (50 μm wide) in the circuit. The LED turned off due to the broken circuit and recovered after 30 min of heating at 105 °C. The healing process is presented in Figure 8a. The electrodes can be used as human finger motion detectors by measuring the changes in capacitance due to bending/stretching. The sensor can distinguish various finger motions due to the fact that each figure motion exhibits a different capacitance value as shown in Figure 8b. The sensor can still retain its original capacitance properties after bending/stretching, demonstrating its excellent mechanical properties. The sensor shows promising use in human motion detection, however the recovery temperature of 105 °C is too high for human interaction and the self-healing time of 30 min is too high for practical uses. The recovery temperature needs to be close to room temperature, such that the device does not harm or cause discomfort to the user. Ideally, the healing time needs to be under one second to ensure safety of the device and reduce lag time. This can be achieved by co-polymerisation, which involves incorporating polymers/hydrogels with low transition temperatures and fast recovery times. Overall, the sensor showed excellent properties including good recyclability, even after the 10th cut—healing process, the modulus recovered to 0.51 MPa (91%) and maximum elongation decreased by only 19% from the original 105%. Self-healable sensors and their characteristics are summarised in Table 4.

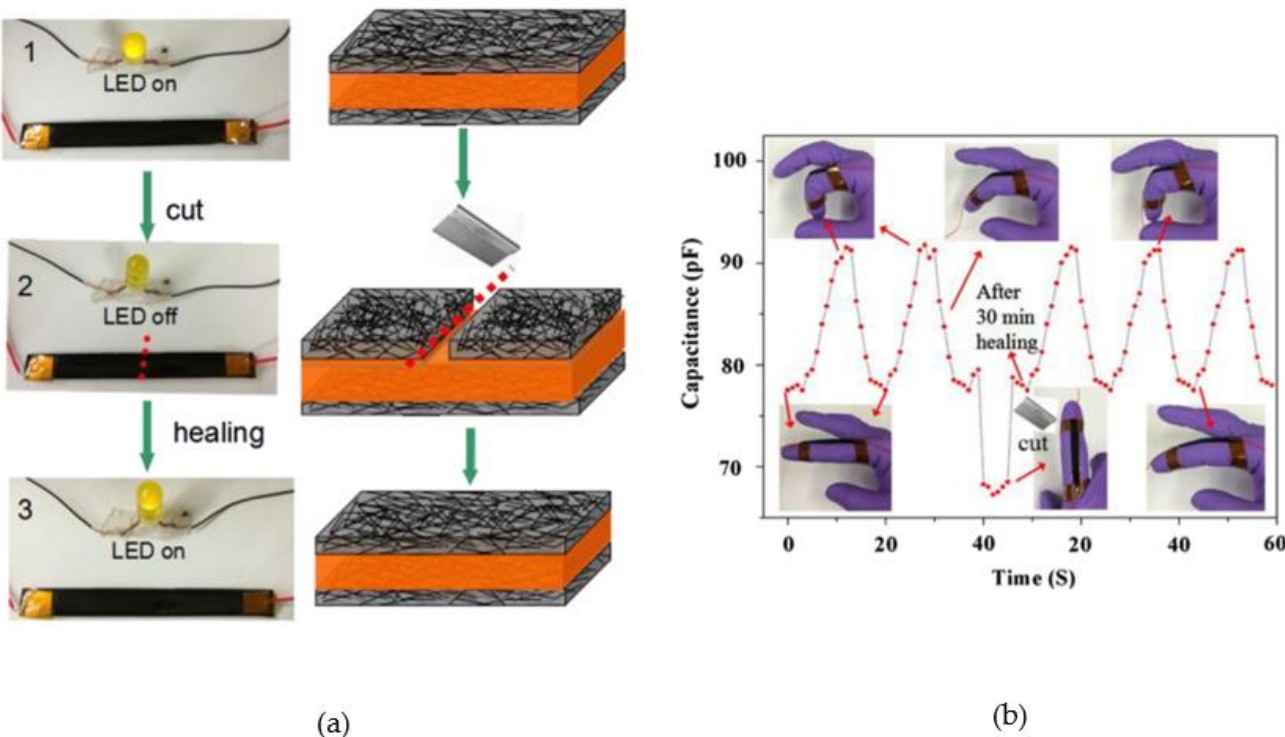

(a)             (b)

**Figure 8.** (**a**) The self-healing process of an electrode based on SWCNT's spread on a flexible, self-healing substrate. Most-top image shows the functional circuit before cutting (LED-ON), middle images show the circuit with the electrode cut (LED-OFF) and the bottom image shows the restored circuit after self-healing process (LED-ON). (**b**) Self-healing finger sensor that can detect various figure motions based on variations in capacitance, reprinted with permission from [95].

**Table 4.** Examples of self-healing sensors and some key characteristics studied in literature.

| Type of Sensor | Materials | Recovery Time | Recovery Temperature/Mechanism |
|---|---|---|---|
| Capacitive sensor. [95] | $CaCu_3Ti_4O_{12}$ and SWCNTs | 30 min | Heating at 150 °C |
| Ammonia gas sensor. [90] | Polyethylenimine (bPEI), polyacrylic acid (PAA), polyethylene terephthalate (PET) and MWCNTs | 30 min | Exposure to DI water |
| Temperature sensor. [84] | Fatty polybasic/diethylenediamine-based oligomers and SWCNTs | 45 min | Heating at room temperature |
| Human–machine interaction/soft robots. [91] | PDMS and silver nanoparticles | ~24 h | Heating at room temperature |
| Human motion detection/electronic skin. [93] | Polyurethane, epoxidized natural rubber and CNTs | 0.06 s | Heating at room temperature |
| Finger motion sensor. [95] | $CaCu_3Ti_4O_{12}$ and SWCNTs | 30 min | Heating at 150 °C |
| Pressure sensor. [97] | PBS/ PDMS and silver microflakes | 6 h | Heating at room temperature |
| Wearable strain sensors. [98] | Nano-chitin, ferric ions, tannic acid and starch/polyvinyl alcohol/polyacrylic acid (St/PVA/PAA) hydrogel | ~60 min | Heating at room temperature |

## 7. Wearables

Wearables have gained much attention in the last decade [11,48,98–104]. These are electronic devices that are attached to the human body, a typical example being smart watches that had an estimated global market share of over 20 billion USD in 2019, 59 billion USD in 2021 and is predicted to rise to 96 billion USD by 2027 [8,9,103,104]. Wearables are improving the way of life by readily providing us with useful information including meteorological data (e.g., humidity and temperature), navigation (e.g., GPS), fitness/exercise (e.g., number of steps per day) and health monitoring (e.g., heart rate, sugar levels and body temperature). A significant amount of wearable research is centred on the relevant material science which includes the mechanical and electrical properties of the sensors [11,12,98,99,101]. Some of the commonly studied characteristics in wearables are summarised in Table 4. The main challenges include enhancing sensitivity, improving comfortability (especially for biomedical wearables), weight reduction, energy consumption optimisation and achieving high stability (e.g., number of bending cycles before fracture, water resistance and thermal stability) [29,99–101]. The cost and mass production of a wearable is heavily dependent on the fabrication process and material cost of the sensors. For example, Liu et al. [99] used electrospinning and acid etching methods to fabricate a heart monitoring wearable which can easily be mass produced due to the low-cost and simplicity of the fabrication method. The device uses piezoelectric technology to self-power itself from the heart vibrations. The research team implemented an innovative method for sensor fabrication that involved placing nano/microscale pores and corona poling in a piezoelectric-based material (polyvinylidene fluoride trifluoroethylene film with ZnO nanoparticles) to improve the sensitivity. The heartbeat sensor could achieve a piezoelectric coefficient $d_{33}$ of 3312 pC/N, a value comparable to commercial sensors. The sensor was compared to a commercial heart sensor during pathological heart sounds testing whereby a speaker was used to produce heartbeat signals while the sensors captured the signal as an input. The output from the electrospun sensor was consistent with the commercial sensor. Another self-powering wearable reported in 2021 [99] has self-healing and stimuli free properties, which the researchers termed SELF (stimuli-free self-healing and self-recovery). The device was fabricated from nano-chitin coated with ferric ions and tannic acid incorporated into a hydrogel (starch/polyvinyl alcohol/polyacrylic acid). The sensor achieved a toughness value of 2.27 $MJ/m^3$ at 15 wt.% nano chitin (dimensions were 10 mm wide, 35 mm long and 6 mm thick).

It is noted that most of the published research is centred on the features and functions including comfortability and durability of the device (e.g., water resistance, thermal resistance and mechanical properties). However, the biocompatibility, toxicity, processability and cost of the wearable strain sensors needs to be examined for industrial applications.

Privacy issues, performance risks and social risks are other noted areas for future investigation [102]. Furthermore, regarding the materials research needs, nanomaterials have gained much attention due to the advanced physicochemical properties and the need to miniaturise wearables [11,41,48,98,99]. Three-dimensional printing is currently and will continue to play an important role in the development of wearables (and sensors in general) due to its ability to customise devices easily, achieve designs not possible via other techniques, reduce or avoid assembly operations, as well as the use of multi-material composites with advanced electrical/mechanical properties [56,59,102]. Wearable sensors and their characteristics are summarised in Table 5.

**Table 5.** Examples of wearables and some key characteristics studied in literature.

| Type of Sensor | Sensitivity (Smallest Detectable Quantity) | Stability/Minimum Bending Cycles | Mean Response Time |
|---|---|---|---|
| Pressure sensor. [12] | >1.14/ kPa | 50,000 bending cycles | 17 ms |
| Respiration monitoring. [15] | 44% relative humidity | 20 bending cycles | 8.9 s |
| Capacitance pressure sensor. [27] | 0.86 pF/bar | n/a | n/a |
| Piezoresistive pressure sensor. [28] | 8.5 mV/bar | 22 bending cycles | 15 s |
| Pressure sensors. [29] | 0.02-0.55 /kPa | n/a | 1 ms–10 s |
| Temperature sensors. [36] | 0.4 mV$^\circ$/C | 600 h | n/a |
| Temperature sensor. [41] | $2.23 \times 10^{-3}$/$^\circ$C | $-269$ and 400 $^\circ$C | n/a |
| Pressure sensors. [47] | 1.80/kPa | 3000 bending cycles | 200 ms |
| Temperature sensor. [48] | 0.214 $\Omega^\circ$/C | 35 to 45 $^\circ$C | 1.6 s |
| Pressure sensor. [50] | 0.7 kPa$^{-1}$ (up to 25kPa) | 100,000 bending cycles | 50 ms |
| Temperature sensor. [50] | 0.83%/K | 22–70 $^\circ$C | 100 ms |
| Piezoelectric-heartrate monitor with self-powering. [99] | 3312 pC/N | 2500 bending cycles | n/a |
| Strain sensing with self-healing. [98] | n/a | 1503% stretchability, 184.1 kPa strength | 60 min (healing time) |

## 8. Conclusions

This review paper highlights the advantages of flexible sensors. These advantages include the ease of fabrication of flexible sensors owing to the advancements in 3D printing methods and the ability to self-heal. Several examples of 3D printing methods that are currently used in manufacturing flexible sensors were discussed, including FDM, FFF, inkjet printing, aerosol jetting, SLM, DIW, SLS and others. Current materials being used in the manufacturing of flexible sensors were discussed, including polymers such as PDMS, PNIPAM, PVP, PET, PEN, PVA and Kapton. Nanomaterials are being incorporated into polymer matrixes to induce electrical properties including nanomaterials of carbon, silver and gold. Hydrogels have also gained much attention in flexible sensor literature, mainly due to their self-healing properties. Self-healing enables the sensor to recover from damages such as cuts and bends incurred during use, and this enables the sensor to have a longer operating life and higher sensitivity. Wearables have become part of our day-to-day life; these include smart watches and biomedical devices that provide real-time health and fitness data.

Flexible sensors have become a topic of interest due to the rising demand in various fields, including stretchable wearables, health monitoring, packaging, soft robotics,

electronic skins and weather monitoring. The most utilised polymer in the fabrication of flexible sensors is PDMS due to its biocompatibility, transparency, non-flammable properties, non-toxicity, self-healing, hydrophobic nature and stretchability. The most used nanomaterials are carbon-based due to the high conductivity and stability of carbon nanomaterials yet being low cost in comparison to silver or gold nanomaterials, which are also being utilised to induce electrical properties in the flexible sensors. FDM, FFF, inkjet printing and direct ink writing are the most used additive manufacturing methods for flexible sensors owing to their low cost, cheapness of materials and ease of fabrication. The main challenge in these printing methods is getting the materials to be in the correct range of rheological properties acceptable by the printer.

**Author Contributions:** Conceptualisation, A.N., M.V. and D.B.; methodology, A.N., M.V. and D.B.; validation, M.V., S.C., B.F. and D.B.; formal analysis, M.V. and D.B; investigation, A.N., M.V., S.C., B.F. and D.B.; resources, M.V., S.C., B.F. and D.B.; data curation, A.N., M.V. and D.B.; writing—original draft preparation, A.N.; writing—review and editing, A.N., M.V., S.C. and D.B.; visualisation, B.F.; supervision, M.V. and D.B.; project administration, M.V. and D.B.; funding acquisition, M.V., B.F. and D.B. All authors have read and agreed to the published version of the manuscript.

**Funding:** This publication has emanated from research supported by a research grant from Science Foundation Ireland (SFI) under grant numbers 18/EPSRC-CDT/3584 and 16/RC/3872 and is co-funded under the European Regional Development Fund.

**Institutional Review Board Statement:** Not applicable.

**Informed Consent Statement:** Not applicable.

**Data Availability Statement:** Not applicable.

**Acknowledgments:** This work is supported by I-Form, the Science Foundation Ireland Research Centre for Advanced Manufacturing. This work is also supported by Advanced Metallic Systems Centre of Doctoral Training (AMSCDT) which incorporates four universities, namely Dublin City University, University College Dublin, The University of Sheffield and The University of Manchester.

**Conflicts of Interest:** The authors declare no conflict of interest.

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
