# Peer review of "Review of Materials and Fabrication Methods for Flexible Nano and Micro-Scale Physical and Chemical Property Sensors"

_applsci, doi:10.3390/app11188563_

Round 1

Reviewer 1 Report

The manuscript is well written, after reading it several times I couldn’t find any major problems in this manuscript, however, before publication, authors are encouraged to address the below comments, minor revisions, to further enhance the readability of the manuscript.

Comments:

  1. The authors have nicely described the summary of mechanisms and current methods in the flexible sensors in the manuscript, if they mention the pros and cons of each sensor shown here, it would be a nice addition to this manuscript.
  2. Page 5, line 166, 4 mM or micromolar is not clear.

Reviewer 2 Report

This article is discussing the review of materials and fabrication methods for flexible Nano and Micro-Scale physical property sensors that highlights the advantages of flexible sensors over rigid sensors.  These advantages include the ease of fabrication of flexible sensors owing to the advancements in 3D printing methods. Several examples of 3D printing methods that are currently used in manufacturing flexible sensors were discussed including FDM, FFF, Inkjet printing, Aerosol jetting, SLM, DIW, SLS, and others.

The content layout is generally smooth and the analysis is complete, the following comments for authors:

Comment 1:

The part of the abstract does not point out the contribution or innovation of this article. It is recommended that the abstract be modified to make the article more readable.

Comment 2:

The font format in the image and text needs to be consistent, for example, the text in Figures 4 and 7 is inconsistent with other pictures.

Comment 3:

The abstract mentioned:

“The most widely used nanomaterials in flexible sensors are carbon and silver, however, other nanomaterials such as iron, copper, manganese dioxide, and gold are also used to provide controlled levels of conductivity or other functional properties.”

However, this article and conclusion did not discuss the applicability of iron, copper, manganese dioxide, and gold.

Especially iron, its conductivity is relatively poor, and the heat generated when the current passes are relatively high. The author lacks discussion in this regard.

Comment 4:

It is recommended to use a table to focus on the comparison of the advantages and disadvantages of various sensors mentioned in the article to increase readability.

Reviewer 3 Report

As mentioned and justified on page 10 in the proposed article, the development of flexible sensors is a subject of major interest and is the subject of a multiplication of scientific articles related to it. Moreover, recent review articles already exist on the subject as also underlined on page 10 (article by Wen et al [3]. From my point of view, a review article should meet one and/or two objectives. It must allow a person outside the field to appropriate the various concepts, operating principles, scientific challenges, etc. related to the subject. It should also provide a clear view of the positioning of current studies in relation to past studies (performance levels, etc.).

In my opinion, the proposed article should be reworked to meet these objectives. The physical principles of the different sensors should be more detailed as it appears for example in reference 3. Considering that recent review articles already exist, it is a pity that this article does not focus more on the very last articles published (44 articles cited were published before 2017 which represents almost 50% of the bibliography used).

I suggest that summary tables or figures be added to have a clear idea of the position of the various sensors (temperature, humidity, pressure, etc.) according to requirements (flexibility, stretchability, durability, sensitivity, transparency, etc.), and concerns, for each targeted application area. Data on conductivity, temperature operating ranges, detection levels (gas concentration, etc), etc. should be provided to take stock of what currently exists. As far as manufacturing techniques are concerned, the advantages and disadvantages should also be summarized in a table.

I suggest to make a specific part on wearable sensors. Then, the section on textile sensors could be expanded by differentiating the different levels of integration (textile = sensor or textile just serving as a substrate for positioning a sensor).

Some paragraphs are placed in specific sections when they are general (e.g. line 225-236 Information placed in the section on pressure sensors but which is also valid for other types of sensors).

Regarding the "self-healing" part, comments on time, temperature, number of cycles should be generalized to all the studies presented as it is done from line 846 to 855.

The title of part 5 should be changed as it is expected that intrinsic conductive polymers (PPy, PANI, etc.) will be presented. The title of Part 5 should be changed as it is expected that intrinsic conductive polymers (PPy, PANI, etc.: what are the results for these different polymers?) Part 5 deals with polymers that can be considered as dielectrics.

Finally, 2 minor remarks:

There is sometimes confusion/mixing between materials and structures (e.g. table 1: nonwoven is a structure and not a material as PDMS can be; silicone (materials) membrane (structure)

The references need to be revised as some information is missing.

Round 2

Reviewer 3 Report

Dear Authors,

I appreciate your answers and the changes made according to my remarks.

Author Response

Thank you for your comments. We have added some statements in the introduction to clarify the novelty of our review paper. We mentioned at least 5 other review papers published in the last 2 years in this field and discussed their strengths and limitations, aiming at showing the uniqueness and the value that our new review brings to the table.

Thank you.